# AutoSlim: Towards One-Shot Architecture Search for Channel Numbers

## Abstract

We study how to set the number of channels in a neural network to achieve better accuracy under constrained resources (*e.g*., FLOPs, latency, memory footprint or model size). A simple and one-shot approach, named AutoSlim, is presented. Instead of training many network samples and searching with reinforcement learning, we train a single slimmable network to approximate the network accuracy of different channel configurations. We then iteratively evaluate the trained slimmable model and greedily slim the layer with minimal accuracy drop. By this single pass, we can obtain the optimized channel configurations under different resource constraints. We present experiments with MobileNet v1, MobileNet v2, ResNet-50 and RL-searched MNasNet on ImageNet classification. We show significant improvements over their default channel configurations. We also achieve better accuracy than recent channel pruning methods and neural architecture search methods with $100\times$ lower search cost.

Notably, by setting optimized channel numbers, our AutoSlim-MobileNet-v2 at 305M FLOPs achieves **74.2% top-1** accuracy, 2.4% better than default MobileNet-v2 (301M FLOPs), and even 0.2% better than RL-searched MNasNet (317M FLOPs). Our AutoSlim-ResNet-50 at 570M FLOPs, without depthwise convolutions, achieves **1.3% better** accuracy than MobileNet-v1 (569M FLOPs).

## 1 Introduction

The channel configuration (*a.k.a*.. filter numbers or channel numbers) of a neural network plays a critical role in its affordability on resource constrained platforms, such as mobile phones, wearables and Internet of Things (IoT) devices. The most common constraints (Liu et al., 2017b; Huang et al., 2017; Wang et al., 2017; Han et al., 2015a), *i.e*., latency, FLOPs and runtime memory footprint, are all bound to the number of channels. For example, in a single convolution or fully-connected layer, the FLOPs (number of Multiply-Adds) increases linearly by the output channels. The memory footprint can also be reduced (Sandler et al., 2018) by reducing the number of channels in bottleneck convolutions for most vision applications (Sandler et al., 2018; Howard et al., 2017; Ma et al., 2018; Zhang et al., 2017b).

Despite its importance, the number of channels has been chosen mostly based on heuristics. LeNet-5 (LeCun et al., 1998) selected 6 channels in its first convolution layer, which is then projected to 16 channels after sub-sampling. AlexNet (Krizhevsky et al., 2012) adopted five convolutions with channels equal to 96, 256, 384, 384 and 256. A commonly used heuristic, the *"half size, double channel"* rule, was introduced in VGG nets (Simonyan & Zisserman, 2014), if not earlier. The rule is that when spatial size of feature map is halved, the number of filters is doubled. This heuristic has been more-or-less used in followup network architecture designs including ResNets (He et al., 2016; Xie et al., 2017), Inception nets (Szegedy et al., 2015; 2016; 2017), MobileNets (Sandler et al., 2018; Howard et al., 2017) and networks for many vision applications. Other heuristics have also been explored. For example, the pyramidal rule (Han et al., 2017; Zhang et al., 2017a) suggested to gradually increase the channels in all convolutions layer by layer, regardless of spatial size. Figure 1 visually summarizes these heuristics for setting channel numbers in a neural network.

Beyond the macro-level heuristics across entire network, recent works (Sandler et al., 2018; He et al., 2016; Zhang et al., 2017a; Tan et al., 2018; Cai et al., 2018) have also digged into channel configuration for micro-level building blocks (a network building block is usually composed of

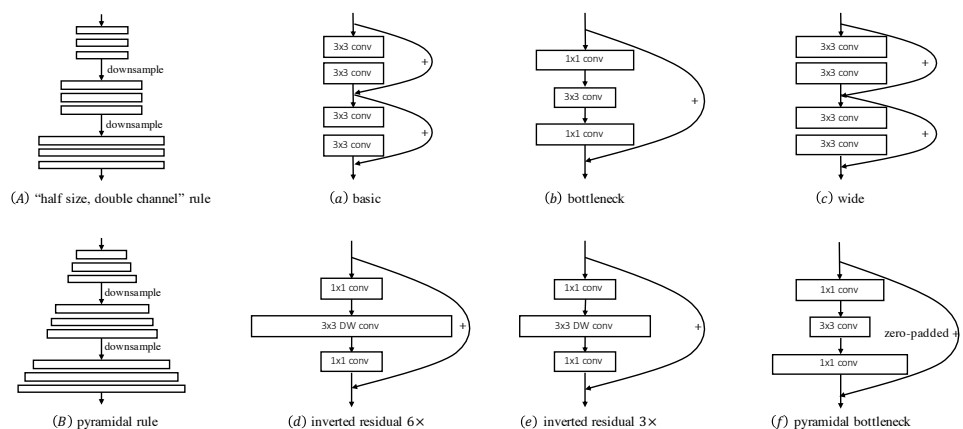

Figure 1: Various heuristics for setting channel numbers across entire network $((A) - (B))$ (Simonyan & Zisserman, 2014; Han et al., 2017; Zhang et al., 2017a), and inside network building blocks $((a) - (f))$ (Sandler et al., 2018; He et al., 2016; Han et al., 2017; Zhang et al., 2017a; Tan et al., 2018; Cai et al., 2018).

several $1 \times 1$ and $3 \times 3$ convolutions). These micro-level heuristics have led to better speed-accuracy trade-offs. The first of its kind, *bottleneck residual block*, was introduced in ResNet (He et al., 2016). It is composed of $1 \times 1$, $3 \times 3$, and $1 \times 1$ convolutions, where the $1 \times 1$ layers are responsible for reducing and then restoring dimensions, leaving the $3 \times 3$ layer a bottleneck ($4\times$ reduction). MobileNet v2 (Sandler et al., 2018), however, argued that the bottleneck design is not efficient and proposed the *inverted residual block* where $1 \times 1$ layers are used for expanding feature first ($6\times$ expansion) and then projecting back after intermediate $3 \times 3$ depthwise convolution. Furthermore, MNasNet (Tan et al., 2018) and ProxylessNAS nets (Cai et al., 2018) included $3\times$ expansion version of *inverted residual block* into search space, and achieved even better accuracy under similar runtime latency.

Apart from these human-designed heuristics, efforts on automatically optimizing channel configuration have been made explicitly or implicitly. A recent work (Liu et al., 2018c) suggested that many network pruning methods (Liu et al., 2017b; Li et al., 2016; Luo et al., 2017; He et al., 2017; Huang & Wang, 2018; Han et al., 2015b) can be thought of as performing network architecture search for channel numbers. Liu *et al.* (Liu et al., 2018c) showed that training these pruned architectures from scratch leads to similar or even better performance than fine-tuning and pruning from a large model. More recently, MNasNet (Tan et al., 2018) proposed to directly search network architectures, including filter sizes, using reinforcement learning algorithms (Schulman et al., 2017; Heess et al., 2017). Although the search is performed on the factorized hierarchical search space, massive network samples and computational cost (Tan et al., 2018) are required for an optimized network architecture.

In this work, we study how to set channel numbers in a neural network to achieve better accuracy under constrained resources. To start, the first and the most brute-force approach came in mind is the exhaustive search: training all possible channel configurations of a deep neural network for full epochs (*e.g.*, MobileNets (Sandler et al., 2018; Howard et al., 2017) are trained for approximately 480 epochs on ImageNet). Then we can simply select the best performers that are qualified for efficiency constraints. However, it is undoubtedly impractical since the cost of this brute-force approach is too high. For example, we consider a $8$-layer convolutional networks and a search space limited to 10 candidates of channel numbers (*e.g.*, 32, 64, ..., 320) for each layer. As a result, there are totally $10^8$ candidate network architectures.

To address this challenge, we present a simple and one-shot solution *AutoSlim*. Our main idea lies in training a slimmable network (Yu et al., 2018) to approximate the network accuracy of different channel configurations. Yu *et al.* (Yu et al., 2018; Yu & Huang, 2019) introduced slimmable networks that can run at arbitrary width with equally or even better performance than same architecture trained individually. Although the original motivation is to provide instant and adaptive accuracy-efficiency trade-offs, we find slimmable networks are especially suitable as benchmark performance estimators for several reasons: (1) Training slimmable models (using *the sandwich rule* (Yu &

Huang, 2019)) is much faster than the brute-force approach. (2) A trained slimmable model can execute at arbitrary width, which can be used to approximate relative performance among different channel configurations. (3) The same trained slimmable model can be applied on search of optimal channels for different resource constraints.

In *AutoSlim*, we first train a slimmable model for a few epochs (*e.g.*, 10% to 20% of full training epochs) to quickly get a benchmark performance estimator. We then iteratively evaluate the trained slimmable model and greedily slim the layer with minimal accuracy drop on validation set (for ImageNet, we randomly hold out $50K$ samples of training set as validation set). After this single pass, we can obtain the optimized channel configurations under different resource constraints (*e.g.*, network FLOPs limited to 150M, 300M and 600M). Finally we train these optimized architectures individually or jointly (as a single slimmable network) for full training epochs. We experiment with various networks including MobileNet v1, MobileNet v2, ResNet-50 and RL-searched MNasNet on the challenging setting of 1000-class ImageNet classification. *AutoSlim* achieves better results (with much lower search cost) compared with three baselines: (1) the default channel configuration of these networks, (2) channel pruning methods on same network architectures (Luo et al., 2017; He et al., 2017; Yang et al., 2018) and (3) reinforcement learning based architecture search methods (He et al., 2018; Tan et al., 2018).

## 2 RELATED WORK

### 2.1 ARCHITECTURE SEARCH FOR CHANNEL NUMBERS

In this part, we mainly discuss previous methods on automatic architecture search for channel numbers. Human-designed heuristics have been introduced in Section 1 and visually summarized in Figure 1.

**Channel Pruning.** Channel pruning (*a.k.a.*, network slimming) methods (Liu et al., 2017b; He et al., 2017; Ye et al., 2018; Huang et al., 2018; Lee et al., 2018) aim at reducing effective channels of a large neural network to speedup its inference. Both training-based, inference-time and initialization-time pruning methods have been proposed (Liu et al., 2017b; He et al., 2017; Ye et al., 2018; Huang et al., 2018; Lee et al., 2018; Frankle & Carbin, 2018) in the literature. Here we selectively review two methods (Liu et al., 2017b; He et al., 2017). He *et al*. (He et al., 2017) proposed an inference-time approach based on an iterative two-step algorithm: the LASSO based channel selection and the least square feature reconstruction. Liu *et al*. (Liu et al., 2017b), on the other hand, trained neural networks with a $\ell_1$ regularization on the scaling factors in batch normalization (BN) (Ioffe & Szegedy, 2015). By pushing the factors towards zero, insignificant channels can be identified and removed. In a recent work (Liu et al., 2018c), Liu *et al*.suggested that many network pruning methods (Liu et al., 2017b; Li et al., 2016; Luo et al., 2017; He et al., 2017; Huang & Wang, 2018; Han et al., 2015b) can be thought of as performing network architecture search for channel numbers. In experiments, Liu *et al*. (Liu et al., 2018c) showed that training these pruned architectures from scratch leads to similar or even better performance than iteratively fine-tuning and pruning a large model. Thus, Liu *et al*. (Liu et al., 2018c) concluded that training a large, over-parameterized model is not necessary to obtain an efficient final model. In our work, we take channel pruning methods (Luo et al., 2017; He et al., 2017; 2018) as one of baselines.

**Neural Architecture Search (NAS).** Recently there has been a growing interest in automating the neural network architecture design (Tan et al., 2018; Cai et al., 2018; Elsken et al., 2018; Bender et al., 2018; Pham et al., 2018; Zoph et al., 2018; Liu et al., 2018a; 2017a; 2018b; Brock et al., 2017). Significant improvements have been achieved by these automatically searched architectures in many vision and language tasks (Zoph et al., 2018; Zoph & Le, 2016). However, most neural architecture search methods (Elsken et al., 2018; Bender et al., 2018; Pham et al., 2018; Zoph et al., 2018; Liu et al., 2018a; 2017a; 2018b; Brock et al., 2017) did not include channel configuration into search space, and instead applied human-designed heuristics. More recently, the RL-based searching algorithms are also applied to prune channels (He et al., 2018) or search for filter numbers (Tan et al., 2018) directly. He *et al*.proposed AutoML for Model Compression (AMC) (He et al., 2018) which leveraged reinforcement learning (deep deterministic policy gradient (Lillicrap et al., 2015)) to provide the model compression policy. MNasNet (Tan et al., 2018) proposed to directly search network architectures, including filter sizes, for mobile devices. In the search, each sampled model is trained on 5 epochs using an aggressive learning rate schedule, and evaluated on a $50K$ validation

set. In total, Tan *et al.*sampled about $8,000$ models during architecture search. Further, Proxyless-NAS (Cai et al., 2018) proposed to directly learn the architectures for large-scale target tasks and target hardware platforms, based on DARTS (Liu et al., 2018b). For each residual block, Proxyless-NAS (Cai et al., 2018) followed the channel configuration of MNasNet (Tan et al., 2018), while inside each block, the choices can be $\times 3$ or $\times 6$ version of *inverted residual blocks*. The memory consumption issue (Cai et al., 2018; Liu et al., 2018b) was addressed by binarizing the architecture parameters and forcing only one path to be active.

## 2.2 SLIMMABLE NETWORKS

Slimmable networks were firstly introduced in (Yu et al., 2018). A general slimmable training algorithm and the switchable batch normalization were introduced to train a single neural network executable at different widths, permitting instant and adaptive accuracy-efficiency trade-offs at runtime. However, one drawback of the switchable batch normalization is that the width can only be chosen from a predefined widths set. The drawback was addressed in (Yu & Huang, 2019), where the authors introduced universally slimmable networks, extending slimmable networks to execute at arbitrary width, and generalizing to networks both with and without batch normalization layers. Meanwhile, two improved training techniques, *the sandwich rule* and *inplace distillation*, were proposed (Yu & Huang, 2019) to enhance training process and boost testing accuracy. Moreover, with the proposed methods, one can train nonuniform universally slimmable networks, where the width ratio is not uniformly applied to all layers. In other words, each layer in a nonuniform universally slimmable network can adjust its number of channels independently during inference. In this work, we simply refer to *nonuniform universally slimmable networks* as slimmable networks, if not explicitly noted. While the original motivation (Yu et al., 2018; Yu & Huang, 2019) of slimmable networks is to provide instant and adaptive accuracy-efficiency trade-offs at runtime for different devices, we present an approach that uses slimmable networks for searching channel configurations of deep neural networks.

## 3 AUTOSLIM: NETWORK SLIMMING BY SLIMMABLE NETWORKS

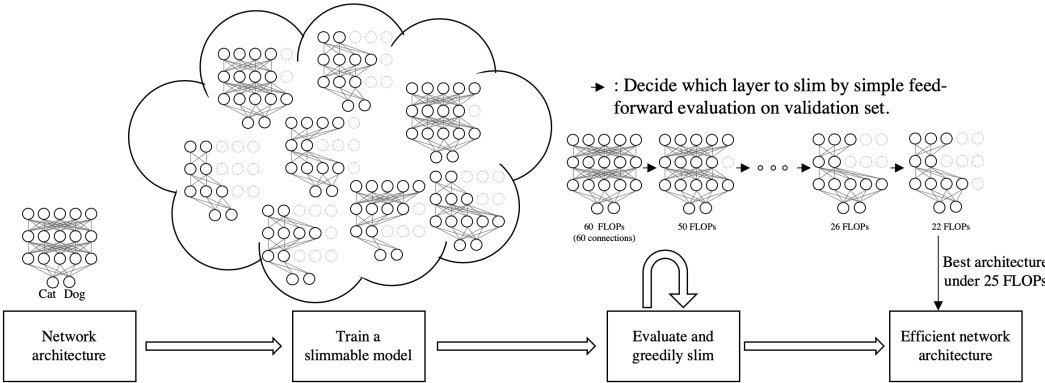

Figure 2: The flow diagram of our proposed approach *AutoSlim*.

In this section, we first present an overview of our proposed approach for searching channel configuration of neural networks. We then discuss and analyze the difference of our approach compared with other baselines, *i.e.*, network pruning methods and network architecture search methods. Afterwards we present each individual module in our proposed solution and discuss its non-trivial details.

## 3.1 OVERVIEW

The goal of channel configuration search is to optimize the number of channels in each layer, such that the network architecture with optimized channel configuration can achieve better accuracy under

constrained resources. The constraints can be FLOPs, latency, memory footprint or model size. Our approach is conceptually simple, and it has two essential steps:

(1) Given a network architecture (*e.g.*, MobileNets, ResNets), we first train a slimmable model for a few epochs (*e.g.*, 10% to 20% of full training epochs). During the training, many different sub-networks with diverse channel configurations have been sampled and trained. Thus, after training one can directly sample its sub-network architectures for instant inference, using the correspondent computational graph and same trained weights.

(2) Next, we iteratively evaluate the trained slimmable model on the validation set. In each iteration, we decide which layer to slim by comparing their feed-forward evaluation accuracy on validation set. We greedily slim the layer with minimal accuracy drop, until reaching the efficiency constraints. No training is required in this step.

The flow diagram of our approach is shown in Figure 2. Our approach is also flexible for different resource constraints, since the FLOPs, latency, memory footprint and model size are all deterministic given a channel configuration and a runtime environment. By a single pass of greedy slimming in step (2), we can obtain the (FLOPs, latency, memory footprint, model size, accuracy) tuples of different channel configurations. It is noteworthy that the latency and accuracy are relative values, since the latency may be different across different hardware and the accuracy can be improved by training the network for full epochs. In the setting of optimizing channel numbers, we benefit from these relative values as performance estimators.

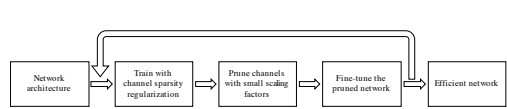

(a) The pipeline of network pruning methods (Liu et al., 2017b).

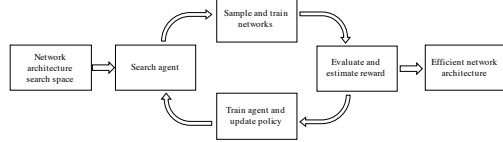

(b) The pipeline of network architecture search methods (Tan et al., 2018; He et al., 2018)

**Discussion.** We compare the flow diagram of our approach with the baselines, *i.e.*, network pruning methods and network architecture search methods.

Many network channel pruning methods (Liu et al., 2017b; Han et al., 2015a; Luo et al., 2017; Han et al., 2015b) follow a typical iterative training-pruning-finetuning pipeline, as shown in Figure 3a. For example, Liu *et al*. (Liu et al., 2017b) trained neural networks with a $\ell_1$ regularization on the scaling factors in batch normalization (BN). After training, the method obtains channels in which many scaling factors are near zero for pruning. Pruning will temporarily lead to accuracy loss, thus the fine-tuning process and a repetitive multi-pass procedure are introduced for enhancement of final accuracy. Compared with our approach, a notable difference is that most network channel pruning methods are grounded on **the importance of trained weights**, thus the slimmed layer usually consists channels of discrete index (*e.g.*, the 4th, 7th, 9th channel are left as important channels while all others are pruned). In our approach, after slimmable training, the importance of the weight is *implicitly ranked by its index*. Thus our approach focuses more on **the importance of channel numbers**, and we always keep the lower-index channels (*e.g.*, all 1st to 3rd channels are left while 4th to 10th channels are slimmed in step (2)). We demonstrate the advantage of our approach by empirical evidences on ImageNet classification with various network architectures.

Network architecture search methods (Tan et al., 2018; Cai et al., 2018; Zoph et al., 2018; Zoph & Le, 2016) commonly consist of three major components: search space, search strategy, and performance estimation strategy. A typical pipeline is shown in Figure 3b. First the search space is defined, based on which the search agent samples network architectures. The architecture is then passed to a performance estimator, which returns rewards (*e.g.*, predictive accuracy after training and/or network runtime latency) to the search agent. In the process, the search agent learns from the repetitive loop to design better network architectures. One major drawback of network architecture search methods is their high computational cost and time cost (Pham et al., 2018; Liu et al., 2018b). Although recently differentiable architecture search methods (Liu et al., 2018b; Luo et al., 2018) were proposed, they cannot be applied on search of channel numbers directly. Most of them (Liu et al., 2018b; Luo et al., 2018) were still using human-designed heuristics for setting channel numbers, which may introduce human bias.

## 3.2 Training Slimmable Networks

**Warmup.** We warmup by a brief review of training techniques for slimmable networks. More details can be found in (Yu et al., 2018; Yu & Huang, 2019). Slimmable networks were firstly introduced and trained with switchable batch normalization (Ioffe & Szegedy, 2015), which employed individual BNs for different sub-networks. During training, features are normalized with current mini-batch mean and variance, thus a simple modification to switchable batch normalization is introduced in (Yu & Huang, 2019): re-calibrating BN statistics after training. With this simple modification, one can train universally slimmable networks (Yu & Huang, 2019) that can run with arbitrary channel numbers. Moreover, two improved training techniques *the sandwich rule* and *in-place distillation* were introduced to enhance training process and boost testing accuracy. We use all these techniques in training slimmable models by default.

**Assumption.** Our approach lies in the assumption that the slimmable model is a good accuracy estimator of individually trained models given same channel configuration. More specifically, we are interested in *the relative ranking of accuracy* among networks with different channel configurations. We use the instant inference accuracy of a slimmable model as the performance estimator. We note that assumptions and approximations commonly exist in other related methods. For example, in network channel pruning methods (Liu et al., 2017b; He et al., 2017), one may assume that weights with smaller norm are less informative and can be pruned, which may not be the case as shown in (Ye et al., 2018). Recently the *Lottery Ticket Hypothesis* (Frankle & Carbin, 2018) was also introduced. In network architecture search methods (Tan et al., 2018; Cai et al., 2018), one may believe the transferability among different datasets, accuracy approximations using aggressive learning rates and fewer training epochs, and approximation in runtime latency modeling.

**The Search Space.** The executable sub-networks in a slimmable model compose the search space of channel configurations given a network architecture. To train a slimmable model, we simply apply two width multipliers (Howard et al., 2017; Yu & Huang, 2019) as the upper bound and lower bound of channel numbers. For example, for all mobile networks (Sandler et al., 2018; Howard et al., 2017; Tan et al., 2018; Cai et al., 2018), we train a slimmable model that can execute between $0.15\times$ and $1.5\times$. In each training iteration, we randomly and independently sample the number of channels in each layer. It is noteworthy that in residual networks, we first sample the channel number of residual identity pathway and then randomly and independently sample channel number inside each residual block. Moreover, we make all layers in a neural network slimmable, including the first convolution layer and last fully-connected layer. In each layer, we divide the channels into groups evenly (*e.g.*, 10 groups) to reduce the search space. In other words, during training or slimming, we sample or remove an entire group, instead of an individual channel. We note that even with channel grouping, the search space is still large. For example in a 10-layer network with 10 channel groups in each layer, the total number of candidate channel configurations is $10^{10}$.

We implement a distributed training framework with synchronized stochastic gradient descent (SGD) on PyTorch (Paszke et al., 2017). We set different random seeds in different processes such that each GPU samples diverse channel configurations in each SGD training step. All other techniques introduced in (Yu et al., 2018) and distributed training techniques introduced in (Goyal et al., 2017) are used by default. All code will be released.

## 3.3 Greedy Slimming

After training a slimmable model, we evaluate it on the validation set (on ImageNet (Deng et al., 2009) we randomly hold out $50K$ images in training set as validation set). We start with the largest model (*e.g.*, $1.5\times$) and compare the network accuracy among the architectures where each layer is slimmed by one channel group. We then greedily slim the layer with minimal accuracy drop. During the iterative slimming, we obtain optimized channel configurations under different resource constraints. We stop until reaching the strictest constraint (*e.g.*, 50M FLOPs or 30ms CPU latency).

**Large Batch Size.** During greedy slimming, no training is involved. Thus we directly put the model in evaluation mode (no gradients are required), which enables us to use a larger batch size (for example during slimming we use mini-batch size 2048 for each GPU with totally 8 V100 GPUs). Large batch size brings two benefits. First, previous work (Yu & Huang, 2019) shows that BN statistics will be accurate if it is calibrated with the batch size larger than $2K$. Thus post-statistics of BN in our greedy slimming can be computed online without additional cost. Second, with large

batch size we can simply use single feed-forward prediction accuracy as the performance estimator. In practice we find it speeds up greedy slimming and simplifies implementation without affecting final performance.

**Training Optimized Networks.** Similar to architecture search methods, after the search, we train these optimized network architectures from scratch. By default we search for the network FLOPs at approximately 200M, 300M and 500M, and train a slimmable model.

## 4 EXPERIMENTS

### 4.1 MAIN RESULTS

Table 1 summarizes our results on ImageNet (Deng et al., 2009) classification with various network architectures including MobileNet v1 (Howard et al., 2017), MobileNet v2 (Sandler et al., 2018), MNasNet (Tan et al., 2018), and one large model ResNet-50 (He et al., 2016). We compare our results with their default channel configurations and recent channel pruning methods (Luo et al., 2017; He et al., 2017; 2018). The top-1 errors of our baselines are from corresponding works (Sandler et al., 2018; Howard et al., 2017; He et al., 2016; Tan et al., 2018; Luo et al., 2017; He et al., 2017; 2018). To have a clear view, we divide the network architectures into four groups, namely, 200M FLOPs, 300M FLOPs, 500M FLOPs and heavy models (basically ResNet-50 based models). We evaluate their latency on same hardware environment with single-core CPU to ensure fairness. Device memory is reported as a summary of all feature maps and weights. We note that the memory footprint can be largely optimized by improving memory reusing and implementation of dedicated operators. For example, the *inverted residual block* can be optimized by splitting channels into groups and performing partial execution for multiple times (Sandler et al., 2018). For all network architectures we train 50 epochs with squeezed learning rate schedule to obtain a slimmable model for greedy slimming. After search, we train the optimized network architectures for full epochs (300 epochs with linearly decaying learning rate for mobile networks, 100 epochs with step learning rate schedule for ResNet-50 based models) with other training settings following previous works (Sandler et al., 2018; Howard et al., 2017; Ma et al., 2018; Zhang et al., 2017b; He et al., 2016; Yu et al., 2018; Yu & Huang, 2019) (weight initialization, weight decay, data augmentation, training/testing image resolution, optimizer, hyper-parameters of batch normalization). We exclude the parameters and FLOPs of Batch Normalization layers (Ioffe & Szegedy, 2015) following common practice since they can be fused into convolution layers.

As shown in Table 1, our models have better top-1 accuracy compared with the default channel configuration of MobileNet v1, MobileNet v2 and ResNet-50 across different computational budgets. We even have improvements over RL-searched MNasNet (Tan et al., 2018), where the filter numbers are already included in its search space. Notably, by setting optimized channel numbers, our AutoSlim-MobileNet-v2 at 305M FLOPs achieves **74.2% top-1** accuracy, 2.4% better than default MobileNet-v2 (301M FLOPs), and even 0.2% better than RL-searched MNasNet (317M FLOPs). Our AutoSlim-ResNet-50 at 570M FLOPs, without depthwise convolutions, achieves **1.3% better** accuracy than MobileNet-v1 (569M FLOPs).

### 4.2 VISUALIZATION AND DISCUSSION

In this part, we visualize our optimized channel configurations and discuss some insights from the results.

**Comparison with Default Channel Numbers.** We first compare our results with default channels in MobileNet v2 (Sandler et al., 2018). We show the optimized number of channels (left) and the percentage compared with default channels (right) in Figure 4. Compared with default MobileNet v2, our optimized configuration has fewer channels in shallow layers and more channels in deep ones.

**Comparison with Width Multiplier Heuristic.** Applying width multiplier (Howard et al., 2017), a global hyper-parameter across all layers, is a commonly used heuristic to trade off between model accuracy and efficiency (Sandler et al., 2018; Howard et al., 2017; Ma et al., 2018; Zhang et al., 2017b). We search optimal channels at 207M, 305M and 505M FLOPs corresponding to MobileNet

Table 1: ImageNet classification results with various network architectures. Blue indicates the network pruning methods (Liu et al., 2018c; Luo et al., 2017; He et al., 2017; 2018; Yang et al., 2018), Cyan indicates the network architecture search methods (Tan et al., 2018; Zoph et al., 2018; Liu et al., 2018a; Zhang et al., 2018) and Red indicates our results using *AutoSlim*.

| Group | Model | Params | Memory | CPU Latency | FLOPs | Top-1 Err. (gain) |
|---|---|---|---|---|---|---|
| | ShuffleNet v1 1.0× | 1.8M | 4.9M | 46ms | 138M | 32.6 |
| | ShuffleNet v2 1.0× | - | - | - | 146M | 30.6 |
| | MobileNet v1 0.5× | 1.3M | 3.8M | 33ms | 150M | 36.7 |
| | MobileNet v2 0.75× | 2.6M | 8.5M | 71ms | 209M | 30.2 |
| 200M FLOPs | AMC-MobileNet v2 | 2.3M | 7.3M | 68ms | 211M | 29.2 (1.0) |
| | MNasNet 0.75× | 3.1M | 7.9M | 65ms | 216M | 28.5 |
| | AutoSlim-MobileNet v1 | 1.9M | 4.2M | 33ms | 150M | 32.1 (4.6) |
| | AutoSlim-MobileNet v2 | 4.1M | 9.1M | 70ms | 207M | 27.0 (3.2) |
| | AutoSlim-MNasNet | 4.0M | 7.5M | 62ms | 217M | 26.8 (1.7) |
| | ShuffleNet v1 1.5× | 3.4M | 8.0M | 60ms | 292M | 28.5 |
| | ShuffleNet v2 1.5× | - | - | - | 299M | 27.4 |
| | MobileNet v1 0.75× | 2.6M | 6.4M | 48ms | 325M | 31.6 |
| | MobileNet v2 1.0× | 3.5M | 10.2M | 81ms | 300M | 28.2 |
| 300M FLOPs | NetAdapt-MobileNet v1 | - | - | - | 285M | 29.9 (1.7) |
| | AMC-MobileNet v1 | 1.8M | 5.6M | 46ms | 285M | 29.5 (2.1) |
| | MNasNet 1.0× | 4.3M | 9.8M | 76ms | 317M | 26.0 |
| | AutoSlim-MobileNet v1 | 4.0M | 6.8M | 43ms | 325M | 28.5 (3.1) |
| | AutoSlim-MobileNet v2 | 5.7M | 10.9M | 77ms | 305M | 25.8 (2.4) |
| | AutoSlim-MNasNet | 6.0M | 10.3M | 71ms | 315M | 25.4 (0.6) |
| | ShuffleNet v1 2.0× | 5.4M | 11.6M | 92ms | 524M | 26.3 |
| | ShuffleNet v2 2.0× | - | - | - | 591M | 25.1 |
| | MobileNet v1 1.0× | 4.2M | 9.3M | 64ms | 569M | 29.1 |
| | MobileNet v2 1.3× | 5.3M | 14.3M | 106ms | 509M | 25.6 |
| | MNasNet 1.3× | 6.8M | 14.2M | 95ms | 535M | 24.5 |
| 500M FLOPs | NASNet-A | - | - | - | 564M | 26.0 |
| | PNASNet-5 | - | - | - | 588M | 25.8 |
| | Graph-HyperNetwork | - | - | - | 569M | 27.0 |
| | AutoSlim-MobileNet v1 | 4.6M | 9.5M | 66ms | 572M | 27.0 (2.1) |
| | AutoSlim-MobileNet v2 | 6.5M | 14.8M | 103ms | 505M | 24.6 (1.0) |
| | AutoSlim-MNasNet | 8.3M | 14.2M | 95ms | 532M | 24.5 |
| | ResNet-50 | 25.5M | 36.6M | 197ms | 4.1G | 23.9 |
| | ResNet-50 0.75× | 14.7M | 23.1M | 133ms | 2.3G | 25.1 |
| | ResNet-50 0.5× | 6.8M | 12.5M | 81ms | 1.1G | 27.9 |
| | ResNet-50 0.25× | 1.9M | 4.8M | 44ms | 278M | 35.0 |
| | He-ResNet-50 | - | - | - | ≈2.0G | 27.2 |
| Heavy Models | | - | - | - | ≈2.9G | 27.0 |
| | ThiNet-ResNet-50 | - | - | - | ≈2.1G | 28.0 |
| | | - | - | - | ≈1.2G | 30.6 |
| | | 23.1M | 32.3M | 165ms | 3.0G | 24.0 |
| | AutoSlim-ResNet-50 | 20.6M | 27.6M | 133ms | 2.0G | 24.4 |
| | | 13.3M | 18.2M | 91ms | 1.0G | 26.0 |
| | | 7.4M | 11.5M | 69ms | 570M | 27.8 |

v2 0.75×, 1.0× and 1.3×. Figure 5a shows the pattern that under different budgets, *AutoSlim* applies different width scaling in each layer.

**Comparison with Model Pruning Methods.** Next, we compare our optimized channel configuration with model pruning method AMC (He et al., 2018). In Figure 5a, we show the number of channels in all layers of optimized MobileNet v2. We observe several characteristics of our op-

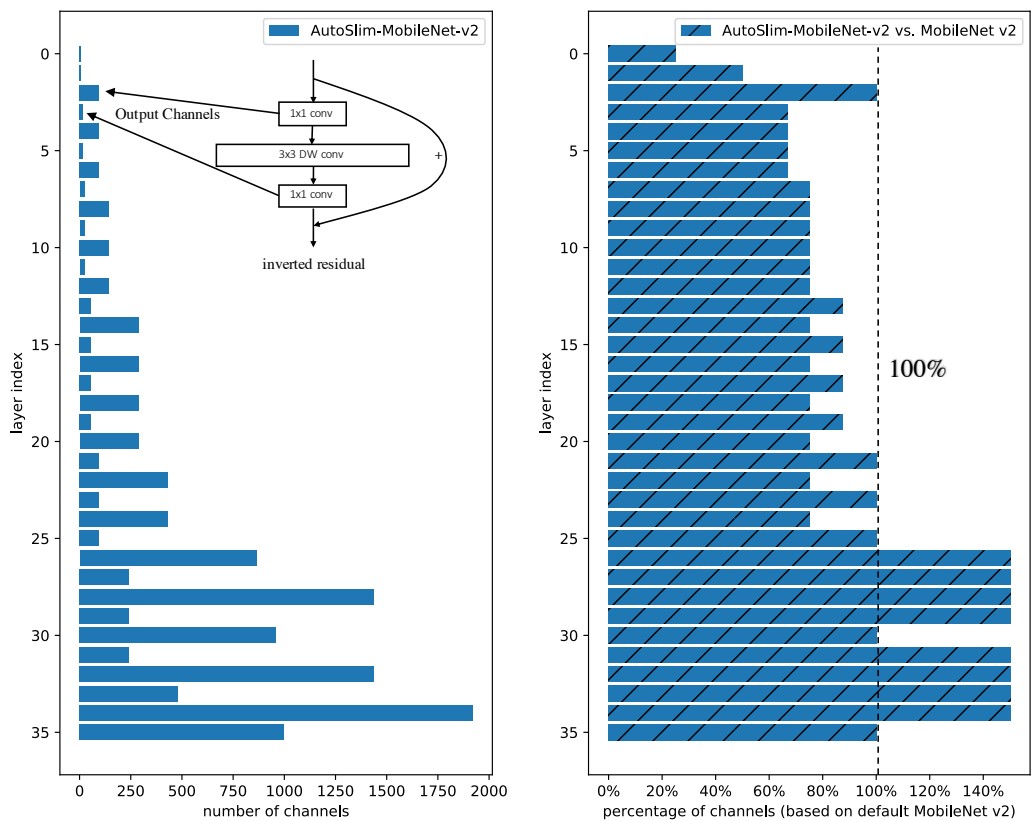

Figure 4: The optimized number of channels (left) and the percentage compared with default channels (right) of MobileNet v2. The channels of depthwise convolutions are ignored in the figure, since its output channels are always equal to the previous $1 \times 1$ convolution outputs.

Table 2: CIFAR10 classification results with default MobileNet v2 and AutoSlim-MobileNet-v2.

| Model | Parameters | FLOPs | Top-1 Err. |
|---|---|---|---|
| MobileNet v2 $1.0\times$ | 2.2M | 88M | 8.1 |
| MobileNet v2 $0.75\times$ | 1.3M | 59M | 8.6 |
| MobileNet v2 $0.5\times$ | 0.7M | 28M | 10.4 |
| AutoSlim-MobileNet v2 | 1.5M | 88M | 6.8 (1.3) |
| AutoSlim-MobileNet v2 | 0.7M | 59M | 7.0 (1.6) |
| AutoSlim-MobileNet v2 | 0.3M | 28M | 8.0 (2.4) |

timized channel configurations. First, AutoSlim-MobileNet-v2 has much more channels in deep layers, especially for deep depthwise convolutions. For example, AutoSlim-MobileNet-v2 has 1920 channels in the second last layer, compared with 848 channels in AMC-MobileNet-v2. Second, AutoSlim-MobileNet-v2 has fewer channels in shallow layers. For example, AutoSlim-MobileNet-v2 has only 8 channels in first convolution layer, while AMC-MobileNet-v2 has 24 channels. It is noteworthy that although shallow layers have a small number of channels, the spatial size of feature maps is large. Thus overall these layers take up large computational overheads.

## 4.3 CIFAR10 EXPERIMENTS

In addition to ImageNet dataset, we also conduct experiments on CIFAR10 (Krizhevsky, 2009) dataset. We use same weight decay hyper-parameter, initial learning rate and learning rate schedule as ImageNet experiments. We note that these training settings may not be optimal for CIFAR10

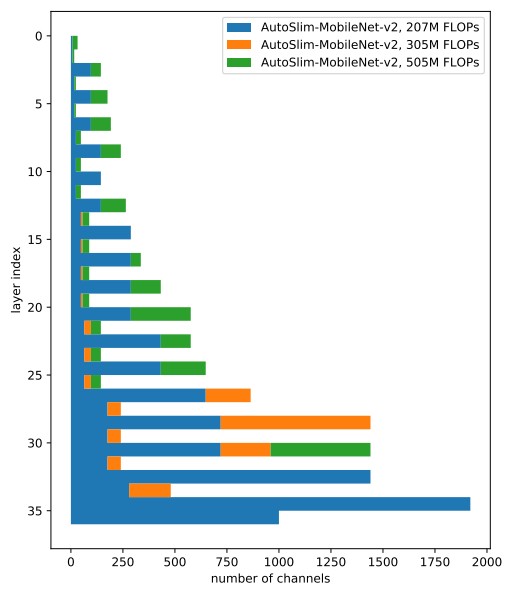 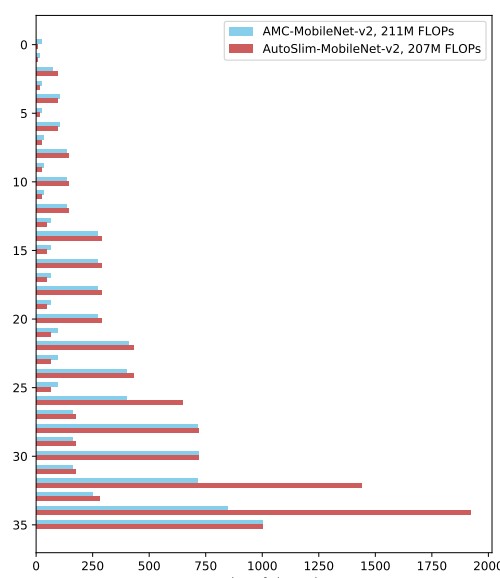

(a) The channels of AutoSlim-MobileNet-v2 at 207M, 305M and 505M FLOPs.

(b) The channels of AutoSlim-MobileNet-v2 compared with AMC-MobileNet-v2.

Table 3: CIFAR10 results with AutoSlim-MobileNet-v2 searched on CIFAR10 or ImageNet.

| Model | Searched On | FLOPs | Top-1 Err. |
|---|---|---|---|
| MobileNet v2 $0.75\times$ | - | 59M | 8.6 |
| AutoSlim-MobileNet v2 | CIFAR10 | 59M | 7.0 $_{(1.6)}$ |
| AutoSlim-MobileNet v2 | ImageNet | 63M | 9.9 $_{(-1.3)}$ |

dataset, nevertheless we report ablative study with same hyper-parameters and settings. We first report the performance of MobileNet v2 (Sandler et al., 2018) with the default channel configurations. We then search with proposed *AutoSlim* to obtain optimized channel configurations at same FLOPs (we hold out $5K$ images from training set as validation set during the search). Finally we train the optimized architectures individually with same settings as the baselines. Table 2 shows that *AutoSlim* models have higher accuracy than baselines on CIFAR10 dataset.

We further study the transferability of the network architectures learned from ImageNet to CIFAR10 dataset, and compare it with the channel configuration searched on CIFAR10 directly. The results are shown in Table 3. It suggests that the optimized channel configuration on ImageNet cannot generalize to CIFAR10. Compared with the optimized architecture for ImageNet, we observed that the optimized architecture for CIFAR10 have much fewer channels in deep layers, which we guess may lead to better generalization on test set for small datasets like CIFAR10. It may also due to the inconsistent image resolutions between ImageNet ($224 \times 224$) and CIFAR10 ($32 \times 32$).

## 5 CONCLUSION

We presented, *AutoSlim*, a simple and one-shot approach on neural architecture search for the number of channels to achieve better accuracy under constrained resources. We demonstrated the effectiveness of *AutoSlim* with extensive experiments on large-scale ImageNet classification and various network backbones including MobileNet v1, MobileNet v2, ResNet-50 and RL-searched MNasNet. *AutoSlim* achieved significant improvements (with much lower search cost) compared with three categories of baselines: the human-designed heuristics, channel pruning methods and architecture search methods based on reinforcement learning. Our proposed solution *AutoSlim* automates the design of channel configurations in a neural network for resource constrained devices.

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
