# OpenReview forum: "AutoSlim: Towards One-Shot Architecture Search for Channel Numbers"
_ICLR.cc/2020/Conference — Reject_

### Official Review · AnonReviewer1 · 2019-10-23
**Official Blind Review #1**

**Rating:** 6

**Review:**

In this paper, the authors propose a method to perform architecture search on the number of channels in convolutional layers. The proposed method, called AutoSlim, is a one-shot approach based on previous work of Slimmable Networks [2,3]. The authors have tested the proposed methods on a variety of architectures on ImageNet dataset.

The paper is well-written and easy to follow. I really appreciate the authors for structuring this paper so well. I have the following questions:

Q1: In figure 4, the authors find that “Compared with default MobileNet v2, our optimized configuration has fewer channels in shallow layers and more channels in deep ones.” This is interesting. Because in network pruning methods, it is found that usually later stages get pruned more [1] (e.g. VGG), indicating that there is more redundancy for deep layers. However, in this case, actually deep layers get more channels than standard models. Is there any justification for this? Is it that more channels in deep layers benefit the accuracy?

Q2: In “Training Optimized Networks”, the authors mentioned that “By default we search for the network FLOPs at approximately 200M, 300M and 500M, and train a slimmable model.” Does this mean that the authors train the final optimized models from scratch as a slimmable network using “sandwich rule” and “in-place distillation” rule? Or are the authors just training the final model with standard training schedule? If it is the first case, can the authors justify why?

Q3: In Table 1, “Heavy Models”, what is the difference between “ResNet-50” and “He-ResNet-50”? Also, why the params, memory and CPU Latency of some networks are omitted?

Q4: In the last paragraph of section 4, the authors tried the transferability of networks learned from ImageNet to CIFAR-10 dataset. I am not sure how the authors transfer the networks from Imagenet to CIFAR-10? Is it the ratio of the number of channels? Can the authors provide the architecture details of MobileNet v2 on CIFAR-10 dataset?

Q5: What is the estimated time for a typical run of AutoSlim? How does it compare to network pruning methods or neural architecture search methods?

Q6: Can the methods be used to search for the number of neurons in fully connected layers? Are there any results?

[1] Rethinking the Value of Network Pruning. Zhuang et al. ICLR 2019
[2] Slimmable neural networks. Yu et al. ICLR 2019.
[3] Universally Slimmable Networks and Improved Training Techniques. Yu et al. Arxiv.


**Experience Assessment:**

I have published one or two papers in this area.

**Review Assessment: Checking Correctness Of Derivations And Theory:**

I carefully checked the derivations and theory.

**Review Assessment: Checking Correctness Of Experiments:**

I carefully checked the experiments.

**Review Assessment: Thoroughness In Paper Reading:**

I read the paper thoroughly.

---

> ### Author Response · Authors · 2019-11-13
> **Authors' Reply to Review**
>
> Thanks for your review efforts! We have addressed all questions below:
>
> Q1: Thanks for delving deep into our discussion in experiments. In our view, the difference of channel number distribution may come from several reasons. First, VGGNet, which many previous pruning methods targeted to optimize, has much more channels than MobileNets designed for efficiency. For example, the last stage of VGG has 4096 channels, ResNet and Inception have 2048 channels, but MobileNet v1 and v2 only have 1024 channels. Different network backbones may lead to the hallucinations like "deeper layers have more redundancy". Second, the channel distribution also depends on the datasets. For example, in Section 4.3 we stated that "we observed that the optimized architecture for CIFAR10 has much fewer channels in deep layers, which we guess may lead to better generalization on test set for small datasets like CIFAR10".
>
> Q2: We trained our final optimized models from scratch as a slimmable network using both “sandwich rule” and “in-place distillation”, as  slimmable models are more flexible to deploy adaptively. Our results are better than the all baseline slimmable models with “sandwich rule” and “in-place distillation” on MobileNet v1, v2 and ResNets [1][2].
>
> Q3: "ResNet-50” means standard ResNets,  “He-ResNet-50" means pruned ResNet-50 [3] by Yihui He et al. Both "He-ResNet-50" and "ThiNet-ResNet-50" did not report their params, memory and CPU Latency, so we omit them. We will make it more clear by inserting references in the table.
>
> Q4: We directly transfer the networks from ImageNet to CIFAR-10 without changing the ratio of the number of channels. We only change the final predictions from 1000 to 10 classes. A similar architecture can be found in Figure 4 (left). Also to transfer, the input resolution is directly changed from 224 (ImageNet) to 32 (CIFAR-10) without modifying other architecture-related configurations.
>
> Q5: Our search cost is mainly on training a slimmable network. In our settings, training a slimmable network for 50 epochs takes roughly the same cost as training a normal network for 200 epochs. Compared with previous architecture search baseline MNasNet, we have more than 100x saving on the search cost (MNasNet samples 8000 models with each model trained with 5 epochs on ImageNet). The strict walltime of the search depends on different platforms and hardware (GPU vs. TPU,  memory and disk, etc.) thus are omitted in our submission.
>
> Q6: Our proposed AutoSlim is a general framework and can indeed be used to search for the number of neurons in fully connected layers. In fact, the final layer in convolutional networks for image classification is a fully-connected layer, and its input number of channels is optimized by AutoSlim.
>
>
> [1] Slimmable neural networks. Yu et al. ICLR 2019.
> [2] Universally Slimmable Networks and Improved Training Techniques. Yu et al. ICCV 2019.
> [3] Channel pruning for accelerating very deep neural networks. Yihui He, Xiangyu Zhang, and Jian Sun. ICCV 2017.

---

### Official Review · AnonReviewer2 · 2019-10-23
**Official Blind Review #2**

**Rating:** 3

**Review:**

This paper proposes a simple and one-shot approach on neural architecture search for the number of channels to achieve better accuracy. Rather than training a lot of network samples, the proposed method trains a single slimmable network to approximate the network accuracy of different channel configurations. The experimental results show that the proposed method achieves better performance than the existing baseline methods.

- It would be better to provide the search cost of the proposed method and the other baseline methods because that is the important metric for neural architecture search methods. As this paper points out that NAS methods are computationally expensive, it would be better to make the efficiency of the proposed method clear.

- According to this paper, the notable difference between the proposed method and the existing pruning methods is that the pruning methods are grounded on the importance of trained weights, but the proposed method focuses more on the importance of channel numbers. It is unclear to me why such a difference is caused by the proposed method, that is, which part of the proposed method causes the difference? And how does the difference affect the final performance?

**Experience Assessment:**

I have read many papers in this area.

**Review Assessment: Checking Correctness Of Derivations And Theory:**

I did not assess the derivations or theory.

**Review Assessment: Checking Correctness Of Experiments:**

I assessed the sensibility of the experiments.

**Review Assessment: Thoroughness In Paper Reading:**

I read the paper at least twice and used my best judgement in assessing the paper.

---

> ### Author Response · Authors · 2019-11-13
> **Authors' Reply to Review**
>
> Thanks for your review efforts! We have addressed all questions below:
>
> 1. Our search cost is mainly on training a slimmable network. In our settings, training a slimmable network for 50 epochs takes roughly the same cost as training a normal network for 200 epochs. Compared with previous architecture search baseline MNasNet, we have more than 100x saving on the search cost (MNasNet samples 8000 models with each model trained with 5 epochs on ImageNet). The strict walltime of the search depends on different platforms and hardware (GPU vs. TPU,  memory and disk, etc.) thus are omitted in our submission.
>
> 2. As suggested in [1], "the pruned architecture itself, rather than a set of inherited important weights, is more crucial to the efficiency in the final model". In our work, the goal is to directly find the best "pruned" architectures. In this context, "the importance of channel numbers" is an explicit and direct optimization objective, while "the importance of trained weights" is just an implicit and surrogate objective. Our final goal of finding efficient network is to find the best architecture, which aligns better with "the importance of channel numbers".
>
>
> [1] Rethinking the Value of Network Pruning. Liu et al. ICLR 2019

---

### Official Review · AnonReviewer3 · 2019-10-27
**Official Blind Review #3**

**Rating:** 3

**Review:**

The paper targets on learning the number of channels across all layers, under computation/model size/memory constraints. The method is simple and the results seems promising.

However, the following issues need to be resolved:
1. The main method is based on published "slimmable networks," such that the novelty is limited;
2. The method is very simpler to DropPath in [1], which uses DropPath to learn important branches while this paper uses it to learn channels. They are similar.
3. Better ablation studies are required in Table 1. This table should be simplified. As the method cannot learn architectures but channel numbers, the only useful pairs of comparisons are those having the same architecture, such as  a pair of MobileNet vs AutoSlim-MobileNet.
4. an important detail is missing: where does the AutoSlim start from? Does it start from a larger model than the baseline? In the set of "500M FLOPs" experiments, I see the size of "AutoSlim-MobileNet v1" (4.6M) is larger than "MobileNet v1 1.0x" (4.2M), this implies that AutoSlim start from a "MobileNet v1 Nx" and N > 1.0. What is exactly N?
5. If AutoSlim starts from a larger baseline model with N times (N > 1.0) width, then the pruning baseline methods (AMC and ThiNet) should also start from the same larger models for fair comparison. In general, starting from a larger model and pruning it down can achieve a better accuracy vs. size trade-off.
6. "300 epochs with linearly decaying learning rate for mobile networks, 100 epochs with step learning rate schedule for ResNet-50 based models", are baselines trained in the same way?

Minor:
1. missing captions in a couple of figures, e.g., Figure 5.
2. "the importance of trained weights" vs "the importance of channel numbers" is trivial


[1] Bender, Gabriel, Pieter-Jan Kindermans, Barret Zoph, Vijay Vasudevan, and Quoc Le. "Understanding and simplifying one-shot architecture search." In International Conference on Machine Learning, pp. 549-558. 2018.

**Experience Assessment:**

I have published in this field for several years.

**Review Assessment: Checking Correctness Of Derivations And Theory:**

N/A

**Review Assessment: Checking Correctness Of Experiments:**

I carefully checked the experiments.

**Review Assessment: Thoroughness In Paper Reading:**

I read the paper thoroughly.

---

> ### Author Response · Authors · 2019-11-13
> **Authors' Reply to Review**
>
> Thanks for your review efforts! We have addressed all questions below:
>
> 1. We hope to reiterate our contribution and novelty in this work. The goal of slimmable networks is to provide adaptive accuracy-efficiency trade-offs and their results are just close to non-slimmable models. Our goal is architecture search on number of channels and our results are much better than baseline models. In this work, we present a one-shot approach for searching number of channels, with extensive results on ImageNet. We proposed the AutoSlim pipeline, designed the search space,  addressed implementation issues and showed better results. Our proposed AutoSlim automates the design of channel configurations for resource constrained devices with much lower search cost.
>
> 2. Thanks for point out the reference. We will discuss this work in our revised version. To the best of our knowledge, the referenced approach has not demonstrated results on searching number of channels, and we are among the first few one-shot approaches on architectural search for number of channels.
>
> 3. Thanks for the suggestion. In Table 1, we think only listing pairs like MobileNet vs AutoSlim-MobileNet is not enough, because the network pruning methods and network architecture search methods are all our baselines and related work. Methods like AMC, NetAdapt and MNasNet all tried to optimize the number of channels of a backbone architecture. We will try to simplify our Table 1 by removing some NAS baselines like NASNet and PNASNet.
>
> 4. We included the detail of AutoSlim in Sec 3.2 The Search Space: "We train a slimmable model that can execute between 0.15x and 1.5x", and Sec 3.3 Greedy Slimming: "We start with the largest model (e.g., 1.5x)". In our search space, N is 1.5.
>
> 5. We start from a larger model (e.g., 1.5x) because we simultaneously compare our discovered models at 1.3x, 1.0x, 0.75x, etc, through a single greedy slimming process. While we agree that pruning baseline methods (AMC and ThiNet) should also start from the same larger models for strictly fair comparison, their search code is not publicly available. Moreover, our discovered small models (e.g., 0.75x) has channels less than 1.0x model in most layers. We believe the performance improvement is mainly from our proposed methods, instead of the search space.
>
> 6. For ResNet-50, the training settings are exactly the same. For mobile networks, we use same settings as ShuffleNets. Official mobile models from google (MobileNets and MNasNets) used a slightly different setting (~380 epochs with exponentially decaying learning rates and RMSProp Optimizer) on TensorFlow. To verify the fairness, we re-implemented and trained MobileNet v2 and MNasNet with strictly same training setting, and the performance is very close. On MobileNet v2 our result is 0.1 better and on MNasNet our result is 0.2 worse.
>
> 7. As suggested in [1], "the pruned architecture itself, rather than a set of inherited important weights, is more crucial to the efficiency in the final model". In our work, the goal is to directly find the best "pruned" architectures. In this context, "the importance of channel numbers" is an explicit and direct optimization objective, while "the importance of trained weights" is just an implicit and surrogate objective. Our final goal of finding efficient network is to find the best architecture, which aligns better with "the importance of channel numbers".
>
>
> [1] Rethinking the Value of Network Pruning. Liu et al. ICLR 2019

---

### Decision · Program_Chairs · 2019-12-19

**Decision:**

Reject

**Comment:**

The paper presents a simple one-shot approach on searching the number of channels for deep convolutional neural networks. It trains a single slimmable network and then iteratively slim and evaluate the model to ensure a minimal accuracy drop. The method is simple and the results are promising.

The main concern for this paper is the limited novelty. This work is based on slimmable network and the iterative slimming process is new, but in some sense similar to DropPath. The rebuttal that PathNet "has not demonstrated results on searching number of channels, and we are among the first few one-shot approaches on architectural search for number of channels" seem weak.